# Deciphering complexity in Pd–catalyzed cross-couplings

George E. Clarke [1,4], James D. Firth [1,4], Lyndsay A. Ledingham[1,4], Chris S. Horbaczewskyj[1], Richard A. Bourne [2], Joshua T. W. Bray [1], Poppy L. Martin[1], Jonathan B. Eastwood[1], Rebecca Campbell[1], Alex Pagett[1], Duncan J. MacQuarrie[1], John M. Slattery [1], Jason M. Lynam [1], Adrian C. Whitwood [1], Jessica Milani[1], Sam Hart [1], Julie Wilson [3] ✉ & Ian J. S. Fairlamb [1] ✉

Understanding complex reaction systems is critical in chemistry. While synthetic methods for selective formation of products are sought after, oftentimes it is the full reaction signature, i.e., complete profile of products/side-products, that informs mechanistic rationale and accelerates discovery chemistry. Here, we report a methodology using high-throughput experimentation and multivariate data analysis to examine the full signature of one of the most complicated chemical reactions catalyzed by palladium known in the chemical literature. A model Pd-catalyzed reaction was selected involving functionalization of 2-bromo-N-phenylbenzamide and multiple bond activation pathways. Principal component analysis, correspondence analysis and heatmaps with hierarchical clustering reveal the factors contributing to the variance in product distributions and show associations between solvents and reaction products. Using robust data from experiments performed with eight solvents, for four different reaction times at five different temperatures, we correlate side-products to a major dominant N-phenyl phenanthridinone product, and many other side products.

Complex reaction networks consisting of hundreds of different species are common in many disparate types of reaction, including oxidation, pyrolysis and polymerization chemistries, with the complexity arising through the presence of many reactants or highly reactive species[1]. In contrast, the complex reaction networks of transition metal-catalyzed processes originate from the nature of catalysis itself, wherein there exists an interplay of distinct elementary steps that must all occur simultaneously within a single environment. Often multiple productive catalytic cycles compete with side reactions and catalyst decomposition pathways. Furthermore, variation in pre-catalyst, ligand, additive, solvent, and reaction stoichiometries, order of mixing and temperature (amongst other variables) can result in dramatic

changes in catalytic speciation and hence reaction outcome(s)[2,3]. Catalyst speciation adds considerable complexity, an aspect that is largely ignored in chemical synthesis, with the focus placed on the product(s) yield[4–6].

Such complex reactions are often met with trepidation by synthetic chemists, with too many variables seemingly outside of their control. Indeed, most reactions under investigation in synthetic chemistry laboratories target selective formation of a single dominant product in high yield. However, there are opportunities to be had through embracing complexity. Firstly, complex reaction manifolds provide an opportunity for serendipitous reaction discovery, which can often lead to the formation of unexpected high-

[1]Department of Chemistry, University of York, Heslington, York YO10 5DD, UK. [2]Institute of Process Research and Development, School of Chemistry & School of Chemical and Process Engineering, University of Leeds, Leeds, LS2 9JT, UK. [3]Department of Mathematics, University of York, Heslington, York YO10 5DD, UK. [4]These authors contributed equally: George E. Clarke, James D. Firth, Lyndsay A. Ledingham. ✉e-mail: julie.wilson@york.ac.uk; ian.fairlamb@york.ac.uk

value products and new reaction chemistries[7,8]. Hartwig and co-workers elegantly demonstrated how complexity could be deliberately employed for discovery. Subjecting complex mixtures of reagents to potential catalysts resulted in the discovery of new reactions[9,10]. Secondly, developing an understanding of a complex reaction and the sensitivities to reaction conditions allows management of side-product profiles, particularly with final product purity and potential toxicity in mind. This is of vital importance in industrial process development where reliability is critical. Thirdly, studying complex reactions can lead to new mechanistic understanding through the identification of competing reaction pathways and interconnected catalytic cycles, that can be used to direct traditional mechanistic studies, i.e., hypotheses built on firm foundations involving real-world substrates and side-product profiles.

The study of complex reactions is no straight-forward task, particularly for those conducting reactions in a traditional manner, one by one[6]. However, high-throughput experimentation (HTE) is perfectly suited for the study of complex transition-metal catalyzed reactions, as it allows the rapid assessment of the effect of reaction conditions on reaction profile, under controlled conditions, whether that be in flow or batch. Indeed, over the past 20 years HTE has become a proven tool for the discovery[8–11], the expansion of and optimization of catalytic transformations in chemical synthesis[12–15].

Herein we report the application of HTE and multi-variate statistical analysis to the study of a complex Pd-mediated transformation. We chose to demonstrate our approach using the Pd-mediated synthesis of N-substituted phenanthridinones 2 from 2-bromo-benzamides 1 (Fig. 1)[16–22]. This transformation is ideal since 1 is pre-configured to be reactively promiscuous, where there are multiple sites with capabilities for C-C and C-N bond formation and a benzamide group that can facilitate multiple bond activations[23]. This unusual reaction has been studied by several research groups and is known to

generate reactive by-products and side-products (vide infra). Crucially for reaction sensitivity assessment, it can be catalyzed by a wide range of catalysts and ligands (including PPh₃, P(furyl)₃, L1, and IPr, Fig. 1), and shows solvent and base-dependent product selectivity[19,20]. The range of Pd pre-catalysts, ligands, bases, solvents, and temperatures (typically run at 100 °C or above) and reaction outcomes observed to date indicate that various catalytically active Pd-species and catalytic cycles are operating, but there is no detailed correlation analysis and conclusions are based on limited empirical evidence.

We investigate this Pd-mediated transformation from the viewpoint of complexity, rather than optimization of a major species. The approach described in this study is intended to provide a strong foundation to help facilitate and direct subsequent downstream mechanism studies on complex reaction systems. This comprehensive study uses a data science approach (Fig. 1) to systematically examine the extent of product and side-product formation in relation to solvent, temperature and reaction time. This approach allows a fuller understanding of the reaction network, providing new insight and advanced chemical knowledge. Moreover, information on the appearance of side products offers opportunities in reaction discovery that could be also useful from a safety perspective.

Further background into the selected reaction (1 → 2): Studies by refs. [19,20] have indicated an understanding behind the formation of phenanthridinones 2 from bromo-benzamides 1 (Fig. 2A, B). Oxidative addition of 1 to a putative Pd⁰ species A was hypothesized to give Pd^II intermediate B (not experimentally evidenced). Subsequent loss of HBr (base-assisted) and second oxidative addition has been proposed to result in the formation of transient Pd^IV species[24] C (based primarily on computational studies using DFT methods, conducted with and without PH₃ as a model phosphine ligand, and no observable transmetallation-type stoichiometric reaction involving Pd^II intermediates)[19]. This goes on to proceed to a key proposed biaryl Pd^II intermediate D (which can be depicted with different Pd coordination modes), following reductive elimination. Subsequent ipso-substitution and elimination of reactive isocyanate and/or aniline and CO₂ affords 7-membered palladacycle E (not experimentally evidenced). Finally, reductive elimination generates phenanthridinone 2.

Several potentially reactive side-products have been identified including ureas 3 (derived from isocyanate and aniline)[16,17] as well as symmetrical biaryls 4[16,20] and amides 5[19,20] and 6[16]. Interestingly, ureas are known to be active ligands[25–28] and reagents[29] in Pd-catalyzed cross-couplings. The relationship and interplay of all these species have not been fully delineated. We recognized that HTE and data analysis of reaction outcomes offers much potential in examining further. It is of particular note that natural[30] and synthetic phenanthridinones[31–36] have been shown to possess a wide range of biological activities, thus a greater understanding of the reaction network would potentially aid synthesis of these useful molecular scaffolds and allow the generation of toxicological profiles for the reaction, in addition to providing a test-bed for the development of complexity embracing experimental methodologies and data analysis[37].

## Results
### Benchmark reaction for deciphering complexity in Pd-catalyzed cross-couplings

We chose to investigate the reaction of 2-bromo-N-phenyl benzamide 1a, primarily because of the potential amide-directed C-H activation of the N-phenyl moiety. Given the ability for a range of Pd pre-catalysts and monodentate ligands to catalyze the formation of phenanthridinones 2 (Fig. 1), we planned to assess the ability of bidentate ligands to affect the transformation, particularly as their characterization in follow-on mechanistic studies (i.e., stoichiometric experiments with Pd) might be more facile. An initial pre-catalyst and ligand screen (see Supplementary Information for full details) showed that 5 mol% Pd(OAc)₂ {formally high purity Pd₃(OAc)₆, nitrite-free}[38] and

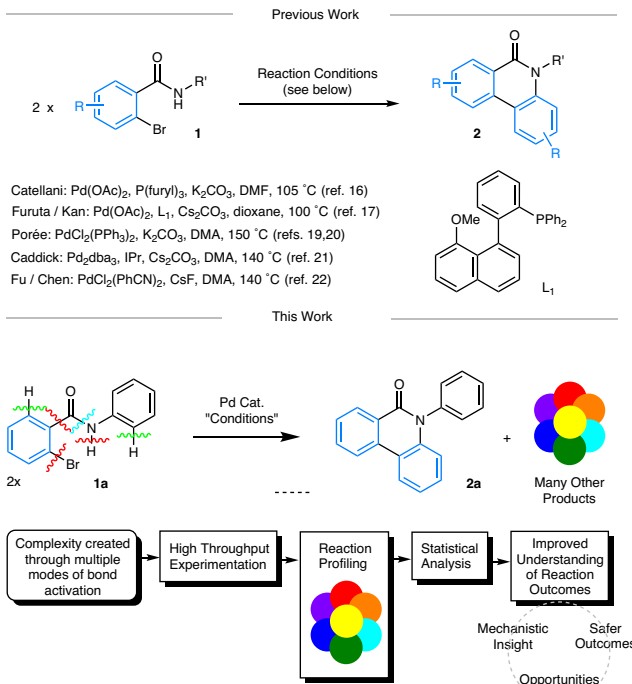

**Fig. 1 | Synthesis of N-substituted phenanthridinones 2 from 2-bromo-benzamides 1—a formal redox-neutral dimerization and deamidation process.** Key: The blue color in structures 1 and 2 shows the origin of the benzo-moiety. The colored disconnection bonds show different bond sites for activation by the catalyst system. Other generic colors are used to guide the eye to multiple products being formed and HTE screening positions in batch mode.

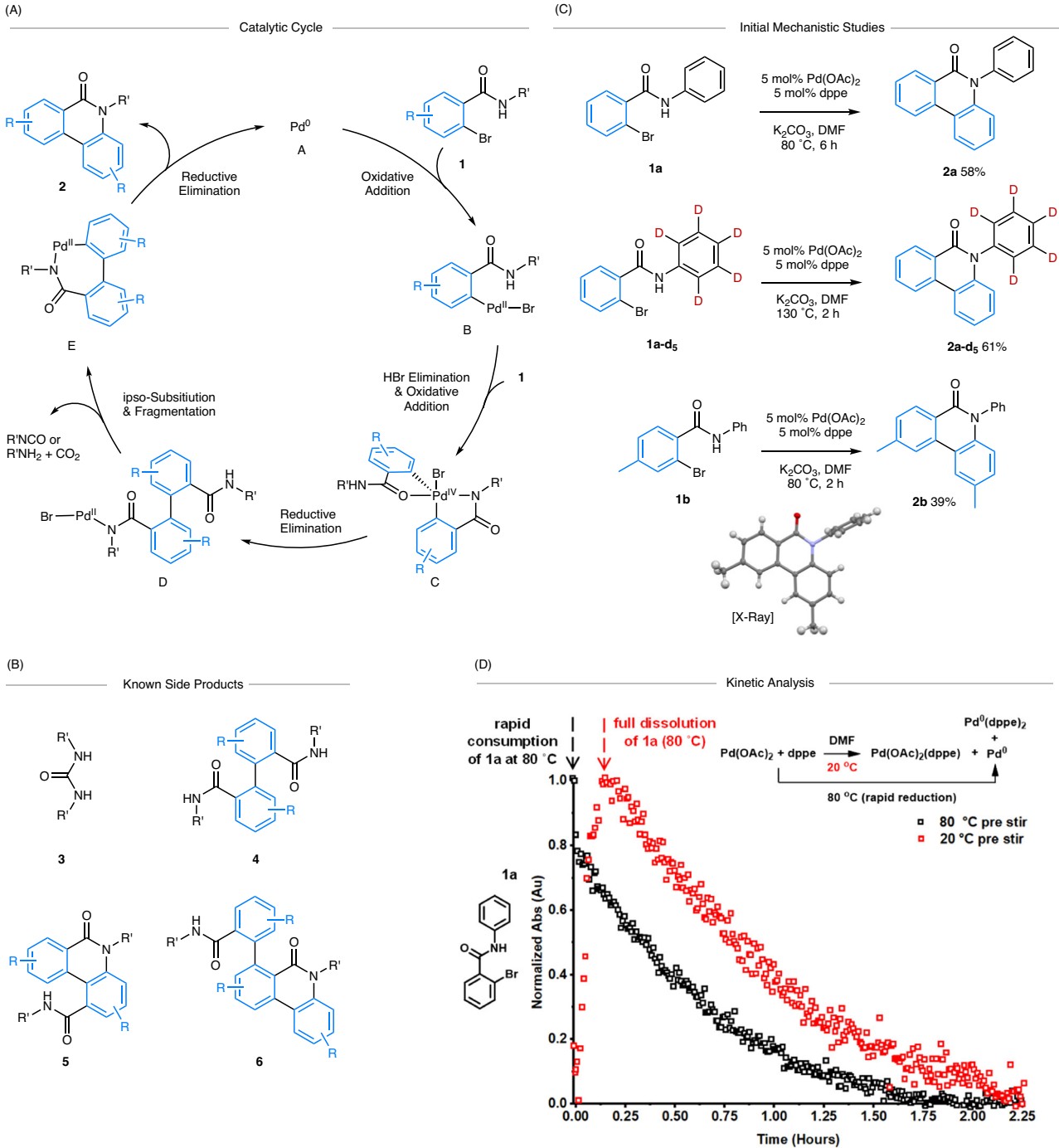

**Fig. 2 | Proposed reaction mechanism for the formation of N-substituted phenanthridinones 2, along with confirmatory studies (this work).** The blue color in compounds shows the origin of the benzo-moiety. **A** Catalytic cycle as proposed by Porée et al.—we note that discrete steps connecting D and E are required[20]. **B** Reported by-products and side-products. Note: that the relationship of PPh₃ to all intermediates is not shown, but it is likely involved in various steps. **C** Synthesis of N-phenyl phenanthridinones **2a**, **2a-d₅** and **2b**. **D** Reaction of **1a** at 80 °C in DMF mediated by Pd(OAc)₂/dppe (1:1, 5 mol%) under two different pre-catalyst regimes. The normalized kinetics show loss of 2-bromo-N-phenylbenzamide **1a** at 1324 cm⁻¹ monitored using a Mettler-Toledo ReactIR/silicon probe. Red square - catalyst pre stir at 20 °C; black square—catalyst pre-stir at 80 °C.

dppe, with K₂CO₃ as base in DMF[39] at 80 °C gave N-phenyl phenanthridinone **2a** in 58% yield (Fig. 2C). Furuta / Kan[17] and Fu / Chen[22] obtained **2a** in 23% and 80% yield respectively from **1a**. We also determined that dppp could be used in place of dppe giving similar results. Crucially, for application to HTE, this reaction was found to be tolerant of air when employing the Pd(OAc)₂/dppe pre-catalyst. Furthermore, conducting the reaction under anhydrous and oxygen-free conditions gave a yield of **2a** which was 54%.

Since the right-hand disubstituted aryl ring in **2a** could be incorporated from either aryl group of **1a** we sought to confirm that it originates from the bromobenzene moiety (rather than the N-phenyl ring). As expected, use of deuterated benzamide **1a-d₅** gave phenanthridinone **2a-d₅** as a single product in 61% yield after 2 h at 130 °C. We further probed the regioselectivity of the reaction by employing methyl-substituted benzamide **1b**, which afforded phenanthridinone **2b** as a single regioisomeric product in 39% yield, the structure of

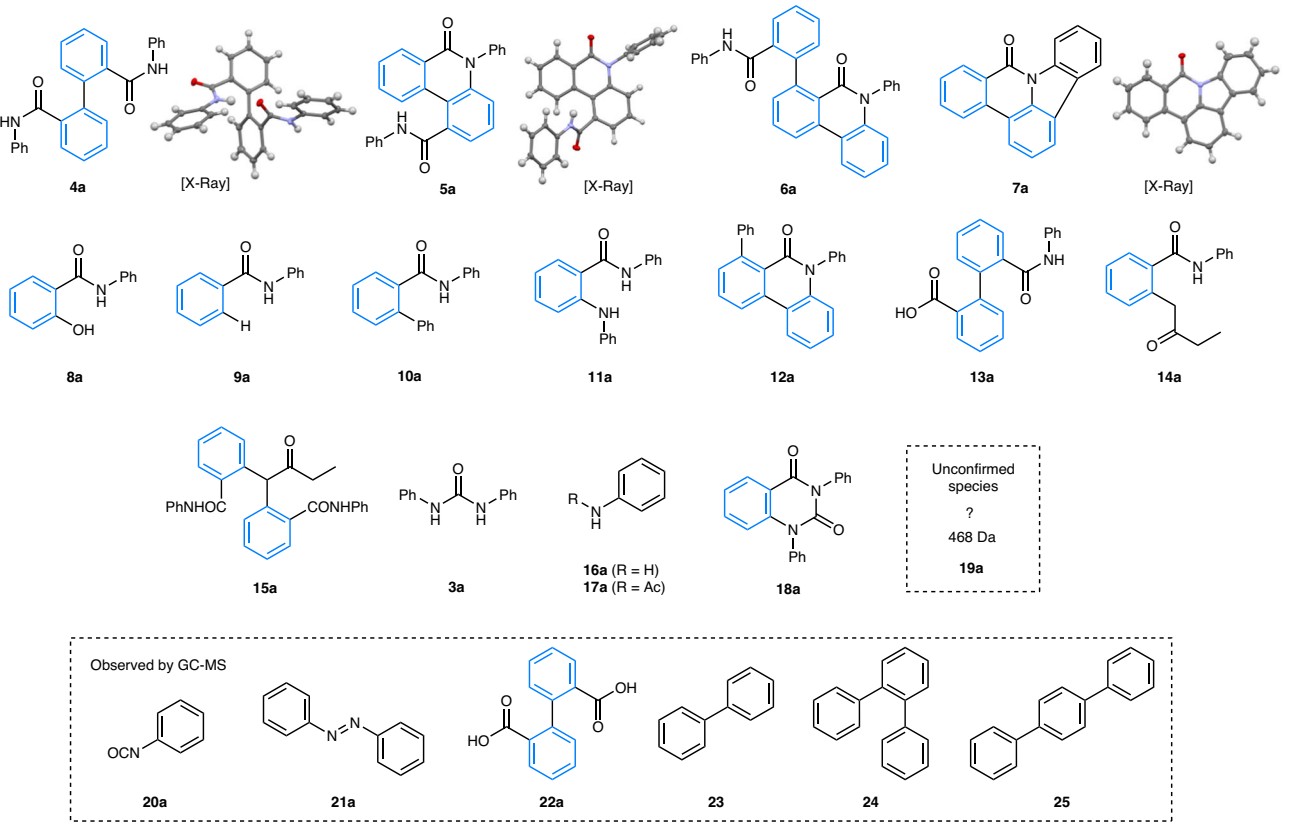

**Fig. 3 | A full survey of the reaction products detected by analytical methods (LC-MS and GC-MS).** The blue color in compounds shows the origin of the benzo-moiety.

which was confirmed by single crystal X-ray diffraction analysis and NMR spectroscopic analysis. This molecular substitution pattern confirms[20] the mechanism proceeds through a biaryl intermediate to **D** (Fig. 2A), under our catalytic reaction conditions.

The reaction of **1a** mediated by Pd(OAc)$_2$/dppe (1:1, 5 mol%) was monitored in operando using IR spectroscopic analysis, with the consumption of **1a** assessed at 80 °C in DMF using both a cold and hot pre-catalyst activation protocol (Fig. 2D). When the reaction mixture was heated from room temperature (*ca.* 20 °C), an induction period of *ca.* 10 min was evident from the color changes recorded (yellow→orange→dark red; the latter indicative of Pd$^0$ species forming, vide infra), with the reaction nearing completion within 2.25 h. The analysis is complicated by the complete dissolution of **1a** requiring heating (which can be seen by the appearance and then subsequent disappearance of **1a**). Heating the pre-catalyst mixture in DMF to 80 °C for 2 min prior to addition of **1a** shortens the induction period considerably (<2 min; reaction mixture a dark red color), with the reaction reaching completion within <2 h (confirmed by independent $^1$H NMR analysis). The difference in the pre-catalyst activation process highlights the impact on the overall reaction time. Independent reactions between Pd(OAc)$_2$ and dppe showed that Pd$^0$(dppe)$_n$ species (n = 1 or 2) were formed (vide infra).

Next, a robust LC-MS method for profiling the reactions from the HTE campaign (in batch mode) was developed. Generally, we observed (by chromatographic and spectroscopic methods) that many products were formed in reaction mixtures accompanying **2a**, **2a-d$_5$**, and **2b**, the majority of which were formed in low amounts but in significant enough quantity to warrant comprehensive profiling. This would fulfill our aspirations to gain greater understanding of this complex reaction network. Characterization of these species was achieved by LC-MS, GC-MS, flash column chromatography and preparative HPLC on the crude reaction mixtures and comparing the spectroscopic and chromatographic data with those of authentic product samples

(see Supplementary Information for full details). Through this approach, we identified a total of 17 side-products and by-products of interest in the LC-MS reaction profiles (Fig. 3), many more than were revealed than in the several previous studies[16–22].

Careful column chromatography resulted in the isolation and characterization of major side-products including symmetrical biaryl **4a**, isolated in ~10% yield (analogous with **4**, Fig. 2B), amide **6a**[16], pentacycle **7a**[40,41], and phenol **8a**. Amides **5a** and **6a** are analogous to compounds **5** and **6** (Fig. 2B) identified by ref. 20 and ref. 16 respectively. Porée et al. showed that phenanthridinones **5** (where R' = benzyl, methyl) were formed in presence of certain base cations and solvents[19,20], for which definitive conclusions could not be drawn. We examined the reaction of **1a** using Porée's pre-catalyst PdCl$_2$(PPh$_3$)$_2$ (5 mol%) with K$_2$CO$_3$ (3 equiv.) in dioxane at 105 °C, however, **5a** was only formed in a trace quantity (by HPLC) in our hands.

Substituted phenol **8a** possibly originates from reductive elimination from a "Ar-Pd-OH" species[42], facilitated by the presence of residual water in the reaction medium. Interestingly, the formation of **8a** indicates the presence of hydroxide in the system, akin to the Suzuki-Miyaura cross-coupling[43,44]. N-phenyl benzamide **9a** likely occurs due to proto-dehalogenation, a side reaction that is common in Pd-catalyzed reactions in basic DMF reaction media, but not fully understood mechanistically[45]. Biaryl amide **10a**, a formal cross-coupled product, was also identified. Interestingly, experiments employing deuterated **1a-d$_5$** and methylated benzamide **1b** starting materials indicated that the "new" aryl ring does not originate from 2-bromo-benzamide **1**, but instead originates from the phosphine ligand (see Supplementary Information for full details). Reactions of phosphine ligands (e.g., PPh$_3$[46] and dppe[47]) with substrates and products at Pd are known, either via liberation of phenyl moieties and phosphido ($^-$PPh$_2$) or phosphonium ($^+$PPh$_4$) ions[2,3].

Other side-products identified include **11a**, presumably formed through a Buchwald-Hartwig amination type reaction of

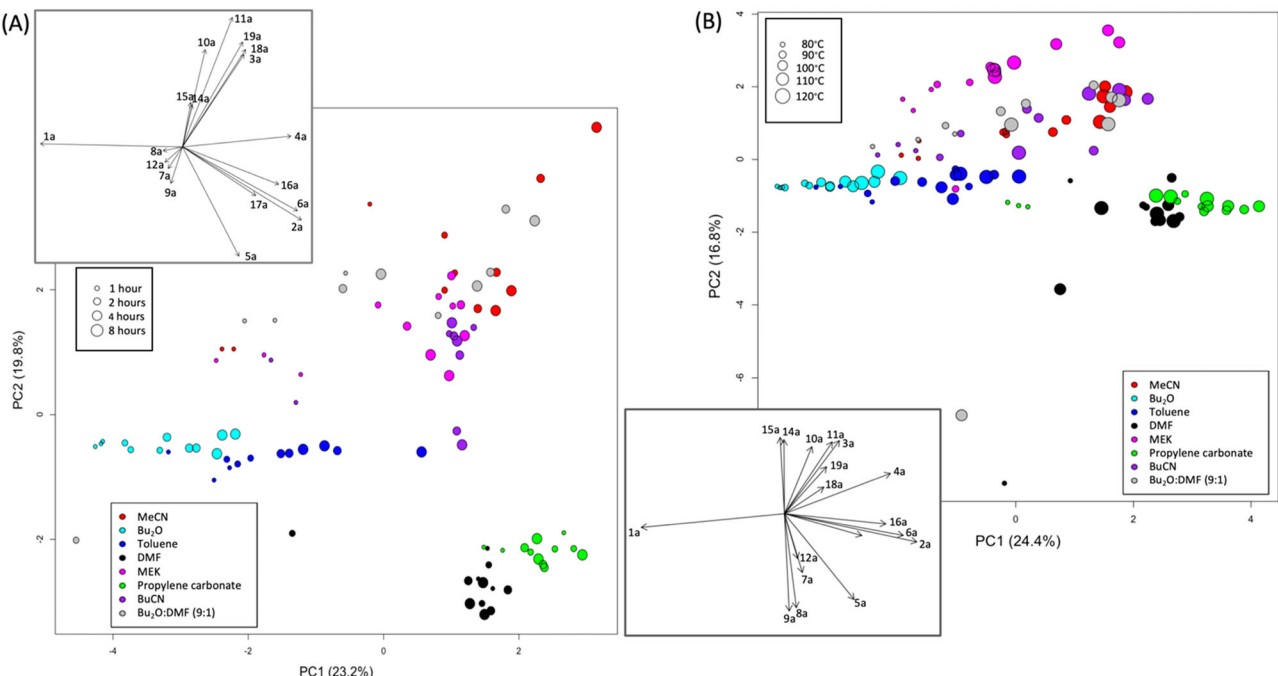

**Fig. 4 | Scores plots for the first two principal components obtained with UV-scaled data from experiments performed in 8 different solvent systems.** The plot in (**A**) shows experiments performed at 110 °C for four different reaction times whereas (**B**) shows 2-h reactions performed at five different temperatures. The loadings, shown as vectors in the insets, indicate the contribution of the various products to the principal components.

2-bromo-N-phenyl benzamide **1a** with aniline **16a**; arylated product **12a**, which presumably arises through amide-directed C-H arylation of phenanthridinone **2a** (as with biaryl **10a**, the "new" aryl ring originates from the dppe ligand); and **13a**, a hydrolysis product of **4a**. Side-products **14a** and **15a** were observed when methyl ethyl ketone (MEK) was used as a solvent and likely arise through the mono- and di-α-arylation of MEK respectively[48–50].

Furthermore, symmetrical urea **3a** was observed by LC-MS along with the requisite building block, aniline **16a** produced during ipso-substitution (**D** to **E**, Fig. 2A)[16,17]. Another aniline derived side-product, acetanilide **17a**, was identified in the analytical LC-MS method. We believe that this is formed by a Pd-mediated acetylation processs[5]. Quinazolinedione **18a**[51] could be formed through a reaction of the substrate **1a** with phenyl isocyanate **20a** (vide infra) and then a subsequent intramolecular amination cyclization process, or a carbonylative-type process.

We observed **19a** by LC-MS, which was present throughout the HTE campaign ($m/z = 469$). However, we were unable to delineate its structure (see Supplementary Information for full details). Thus, we have treated **19a** as being an unknown species. As we demonstrate below, our approach to complex reaction analysis allows the effects of changing reaction parameters on this unknown species / contaminant to be uncovered.

Finally, we identified several species by GC-MS of the reaction mixtures that were invisible to the LC-MS method (Fig. 3). These include the expected by-product phenyl isocyanate **20a**, azobenzene **21a** from oxidative coupling of aniline, which is likely promoted by Pd nanoparticles[52], hydrolysis product **22a** and bi- and terphenyls **23, 24,** and **25**. Whilst these species were not included in the profiling of reaction from our HTE work vide infra, their presence confirms the high degree of complexity of the reaction under study. Moreover, the identification of these species via GC-MS, but not LC-MS highlights that the sole use of one analysis method (which is commonplace in HTE reaction screening) ought to be viewed with some caution.

With a useful LC-MS method established, we next explored the effect of changing reaction conditions on the reaction profile using

HTE. Given that most reported examples of phenanthridinone **2** syntheses use polar aprotic solvents (typically DMF and DMA) at temperatures between 100–150 °C (Fig. 1), alongside the solvent effects observed by refs. 19,20, we selected to study the effect of solvent and temperature in further detail. To this end we selected seven separate solvents {DMF, propylene carbonate, acetonitrile, n-butyronitrile, methyl ethyl ketone (MEK), di-n-butyl ether and toluene} that covered a wide range of polarities, as well as a 9:1 n-Bu₂O:DMF mixture (the latter solvent mixture to aid greater solubility of the reaction components). Furthermore, propylene carbonate[53,54], and MEK[55] were selected primarily as potentially greener and less toxic alternatives. Five temperatures between 80–120 °C (10 °C intervals) were chosen, as were several reaction time-points from 1 to 8 h to provide a temporal visualization of the altering product(s) profile. Reactions were conducted at 130 °C, but significant solvent losses were noted in these experiments leading to unreliable data. We elected to perform the HTE campaign without a catalyst pre-activation step (see Fig. 2D) to avoid variability in the experimental workflow, which facilitated the range of side-products generated to be fully explored.

Reactions were performed on a Chemspeed ISYNTH robotic platform with a solid-dispensing unit[14,15,56–58] to expedite reaction set-up and sampling and off-line LC-MS analysis was used to generate the reaction profile (see Supplementary Information for full experimental details, including workflow schematic diagram). We rigorously assessed positional variability and reproducibility using the ISYNTH system. In total, 40 reactions were performed in triplicate, with sampling at four different time points, generating 480 reaction profiles. LC-MS alignment and peak picking were performed using automated processing (Progenesis QI)[59] and relative concentrations of the reaction constituents were obtained by normalizing the mass-ion counts over the peaks of interest (see Supplementary Information for full details), resulting in a semi-quantification of the species of interest. This approach allows the study of the variation in amount of these (characterized and uncharacterized) species with changing reaction conditions and is simpler than developing an analytical method to show absolute quantities of all species (which would be a

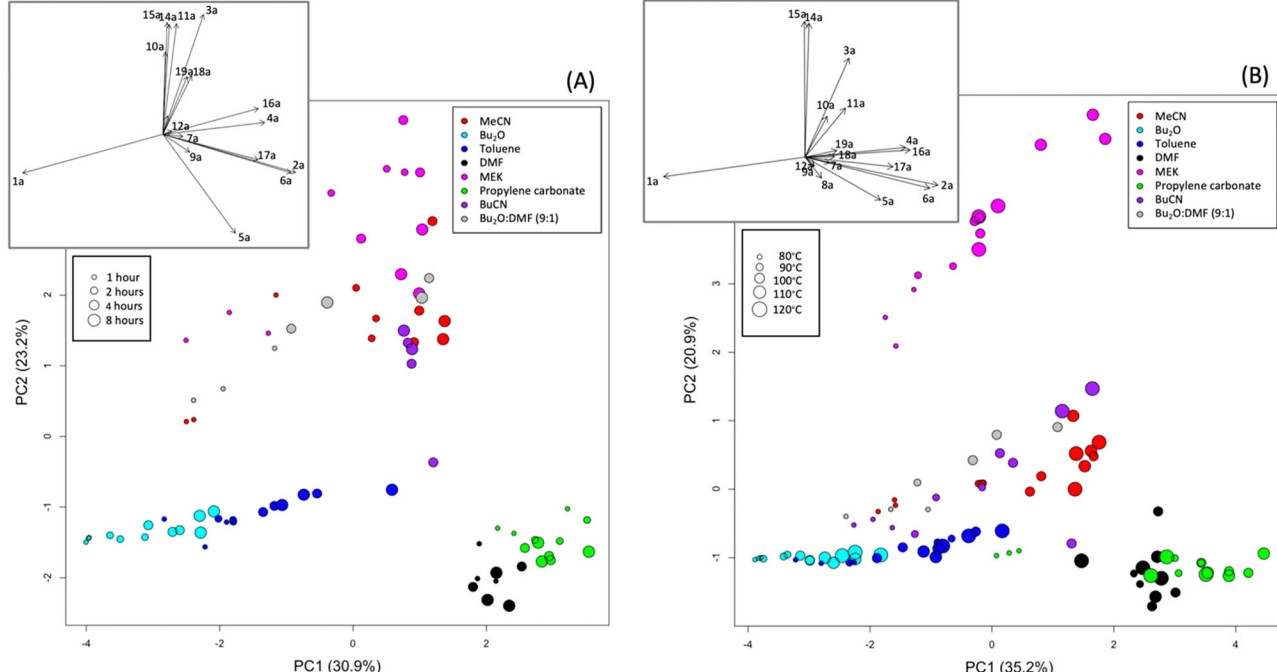

**Fig. 5 | Scores plots for the first two principal components obtained after removing outliers from UV-scaled data from experiments performed in 8 different solvent systems.** The plot in (**A**) shows experiments performed at 110 °C for four different reaction times whereas (**B**) shows 2-h reactions performed at five different temperatures. In both cases, 13 observations were removed. The loadings, shown as vectors in the insets, indicate the contribution of the various products to the principal components.

non-viable effort in most academic/industrial settings for reactions of this complexity).

### Data analysis of reaction outcomes

Analysis of such large multidimensional datasets is difficult without multi-variate statistical analysis techniques to reduce dimensionality. To visualize the effects of reaction variables, we employed principal component analysis (PCA)[60], an unsupervised data analysis method that allows patterns in data with multiple variables to be observed in scatter plots with minimal loss of information. This is achieved by a rotation of the multi-dimensional axes, where each axis corresponds to a different variable, in our case the integrated intensities of identified peaks from LC-MS analysis. The rotation preserves the orthogonality of the axes but the first new axis, or principal component, is chosen as the direction of maximum variance in the data, the second principal component corresponds to the next most variance (orthogonal to the first), and so on. In this way, a small subset of the new axes, or principal components, provides most of the information in the data and pairwise scatter plots showing the coordinates of the observations in relation to even just the first two principal components can reveal meaningful patterns in the data. As these new axes are obtained by a rotation, each principal component is a linear combination of the original variables or peak intensities, and the rotation matrix provides the coefficients, or loadings, of each in the linear combinations. The magnitudes of the loadings for a particular component therefore show the importance of each species to any patterns observed in the direction of that component.

We first examined data obtained at a single temperature to reduce complexity and allow the effect of time and solvent to be studied independently of temperature. At 110 °C these effects are pronounced and highlight important trends. Indeed, PCA shows clear differences between solvents (Fig. 4). For PCA plots including all times and temperatures see the Supplementary Information. The insets show the loadings for the first two principal components (PC1 and PC2) as

vectors, describing the contribution of the different compounds to the patterns seen in the corresponding scores plots.

As the variance for large peaks is greater than that for small peaks, the major products dominate the analysis, and differences due to small peaks are masked unless the variables are rescaled. UV-scaling, or scaling to unit variance gives all variables equal influence on the analysis. PCA scores plots for unscaled data are shown in the Supplementary Information. Here, differences along PC1, accounting for 97.3% of the variance in the data, are related to differences between 2-bromo-benzamide **1a**, the starting material, and major product **2a**. As the variance involves squared values, the scores plot can be mirrored along either PC1 or PC2 and needs to be oriented by looking at the data used in the analysis (provided as supplementary material). In this case, the data shows that Bu₂O observations have the highest values of 2-bromo-benzamide **1a** remaining and MeCN observations have the highest values of product **4a**. Thus, positive PC1 scores are associated with high amounts of **2a** and negative scores with high amounts of **1a**, whilst positive PC2 scores are associated with the higher levels of other products.

After scaling the data so the analysis is not dominated by the large amounts of 2-bromo-benzamide **1a** and phenanthridinone **2a**, PCA shows the distribution of side-products associated with different solvents (Fig. 4). After orienting the scores plot using the original data, it can be seen that, while DMF and propylene carbonate gave the greatest amount of **2a**, regardless of reaction time, Bu₂O led to the most **1a** remaining, with a slow but steady increase in product **2a** with increasing reaction time (Fig. 4A). Other solvents, notably toluene, show a dependence on reaction time with most 1-h reactions in the center of the plot. Bu₂O and toluene have the lowest values of most products, except perhaps products **7a, 8a, 9a** and **12a**. The large positive loadings along PC1 for the major product **2a**, as well as aniline **16a**, and amides **5a** and **6a** show that DMF and propylene carbonate observations on the right of the plot are associated with greater quantities of these products. In fact, further analysis (see Fig. 5) shows that, while both solvents produce high amounts of the major product

**Table 1 | The range of values for each compound. Values below the low/medium threshold were considered "low" and values above the medium/high threshold were considered "high" whilst values between the two thresholds were considered "medium"**

|      | Minimum value | Low/medium threshold | Medium/high threshold | Maximum value |
|------|---------------|----------------------|-----------------------|---------------|
| X1a  | 0.77          | 32.96                | 65.16                 | 97.35         |
| X2a  | 0.48          | 22.39                | 44.29                 | 66.2          |
| X3a  | 0.06          | 3.9                  | 7.74                  | 11.57         |
| X4a  | 0.69          | 11.56                | 22.43                 | 33.3          |
| X5a  | 0.02          | 1.74                 | 3.45                  | 5.16          |
| X6a  | 0             | 1.53                 | 3.05                  | 4.58          |
| X7a  | 0.02          | 3.44                 | 6.87                  | 10.29         |
| X8a  | 0.04          | 0.35                 | 0.67                  | 0.98          |
| X9a  | 0.02          | 0.43                 | 0.84                  | 1.25          |
| X10a | 0.36          | 4.23                 | 8.1                   | 11.97         |
| X11a | 0.16          | 5.1                  | 10.04                 | 14.98         |
| X12a | 0.01          | 2.23                 | 4.45                  | 6.67          |
| X14a | 0             | 6.23                 | 12.46                 | 18.69         |
| X15a | 0             | 0.42                 | 0.84                  | 1.26          |
| X16a | 0.01          | 0.47                 | 0.94                  | 1.4           |
| X17a | 0             | 0.61                 | 1.21                  | 1.82          |
| X18a | 0.07          | 1.37                 | 2.67                  | 3.97          |
| X19a | 0             | 0.26                 | 0.52                  | 0.78          |

**2a**, propylene carbonate produces higher quantities of **6a** whereas DMF is more associated, with higher quantities of **5a** and **16a**.

When considering only 2-h reaction time points performed at different temperatures, PCA reveals similar patterns with solvent (Fig. 4B). Again, reactions using Bu₂O have most **1a** remaining. Except for the lowest temperature reactions (reactions at 80 °C in DMF and 110 °C in 9:1 n-Bu₂O/DMF), DMF and propylene carbonate result in the most phenanthridinone **2a**. Most solvents show higher conversion of **1a** with an increase in temperature. The greater quantities of side-products in MeCN and MEK can be seen again but outcome for both BuCN and the dual solvent system is variable. In addition to product **2a**, higher quantities of aniline **16a** and **6a** could be associated with DMF and propylene carbonate, whilst Bu₂O and toluene have the lowest values. Much of the variance along PC2 is due to unusually high proportions of some compounds being recorded for a few observations, emphasized by the scaling. As the replicate experiments did not have similar outcomes, these unusual observations could be considered outliers due to reaction sampling or analysis errors.

To show any trends with reaction time or temperature more clearly, outliers were removed, and the analysis repeated. Outliers were determined by considering the similarity of replicates. For each set of replicates, the Euclidean distance to the centroid was calculated. The mean distance plus 1.5 standard deviations was set as a threshold and any replicates with a distance greater than this were removed. This threshold was chosen as it removed the worst outliers from each dataset without taking out too many observations. Figure 5 shows the results for scaled data. The reaction time data now shows that MEK is most associated with greater amounts of a combination of products **10a, 11a, 14a, 15a** and **3a**, while MeCN, BuCN and the dual solvent system are more related with greater amounts of products **18a** and **19a** (Fig. 5A). The temperature data set shows an even greater difference in the amount of these side-products for MEK and the increasing trend with temperature can be seen more clearly for MeCN, BuCN and the dual solvent system.

Although PCA gives a good indication of the products associated with the different solvents, interpretation is complicated when several variables have similar loadings. Importantly, it is the sum of the variables that contributes to the principal component in question. For example, a lower value for side-product **6a** may be compensated for by a higher value for aniline **16a**, leading to a similar score as a higher value for **6a** with a lower value for **16a** (Fig. 4B). To determine more specific relationships between solvents and reaction products, we employed correspondence analysis (CA)[33]. The median quantities were calculated over replicate reactions and, for each solvent, the number of observations counted with high, medium, and low quantities of each product, overall temperatures and reaction times (a total of 20 for each solvent). Quantities less than a third of the full range (i.e., the maximum minus minimum value over all solvents) for a product were defined as low, quantities between a third and two-thirds as medium, with quantities above two-thirds of the range for the product considered as high (see Table 1).

The results are shown in Fig. 6A. The strength of the association between solvents and reaction products depends on the distance from the origin (where the dotted lines cross) and on the angle between the vectors from the origin to the points representing the solvents and products. As in PCA, the association of DMF and propylene carbonate with high amounts of the phenanthridinone product **2a** can be seen, but here acetanilide **17a** is associated with propylene carbonate, whilst **5a** and aniline **16a** are associated with DMF. Furthermore, **6a** is associated with both propylene carbonate and, to a lesser extent, DMF.

The biplot also shows that MeCN is related to relatively high amounts of Buchwald-Hartwig product **11a**, whilst MEK is most associated with MEK α-arylation products **14a** and **15a** (as expected for the latter solvent which serves as a substrate). Other solvents, being closer to the origin, are less well discriminated.

The bubble plot in Fig. 6B provides another means for data visualization. Here, only the number of experiments with high quantities are included. It can be readily seen that product **5a** is associated with reaction in DMF with 16 of the 20 reaction profiles recorded when DMF was used (5 temperatures for 4 reaction times) show relatively high quantities of **5a**. Conversely, when the reaction is performed in propylene carbonate, no reactions exhibit high quantities of **5a**. However, 14 of the 20 reactions show high levels of **6a**. Additionally, 12 reactions show relatively high amounts of acetanilide **17a**. Again, that high amounts of α-arylation products **14a** and **15a** are only associated with MEK, which is as expected, validating our approach.

## Heatmap analysis

To garner further mechanistic insight, we used heatmaps to visualize correlations between products. The heatmap in Fig. 7 shows the strength of any correlation across all reactions between the various products. Therefore, the correlation of a product with itself along the diagonal has the strongest positive correlation. The data includes reaction solvents, temperatures, and times. For example, the similarity of MEK α-arylation products **14a** and **15a** is clear from the dark red block indicating a positive correlation close to 1 (highlighted by a yellow box). The strongest negative correlations (dark blue) are between 2-bromo-benzamide **1a**, and the block of positively correlated products, aniline **16a**, biaryl diamide **4a**, **2a**-derivatized compound **6a,** and phenanthridinone **2a** (the major product of the reaction). This negative correlation highlights that significant consumption of **1a** is correlated with formation of phenanthridinone **2a** and the most prevalent side products **4a** and **6a**.

Aniline **16a** correlates with carboxylic acid **13a** (with both possibly formed from the hydrolysis of **4a**) and **5a**. Oxidative reaction of **2a** (by C-H bond activation) with urea **3a** could in principle lead to both **5a** and **16a**, leading to positive correlations.

Other blocks of positively correlated products show **7a** (oxidative cyclization product) and **9a** (reductive proto-debrominated

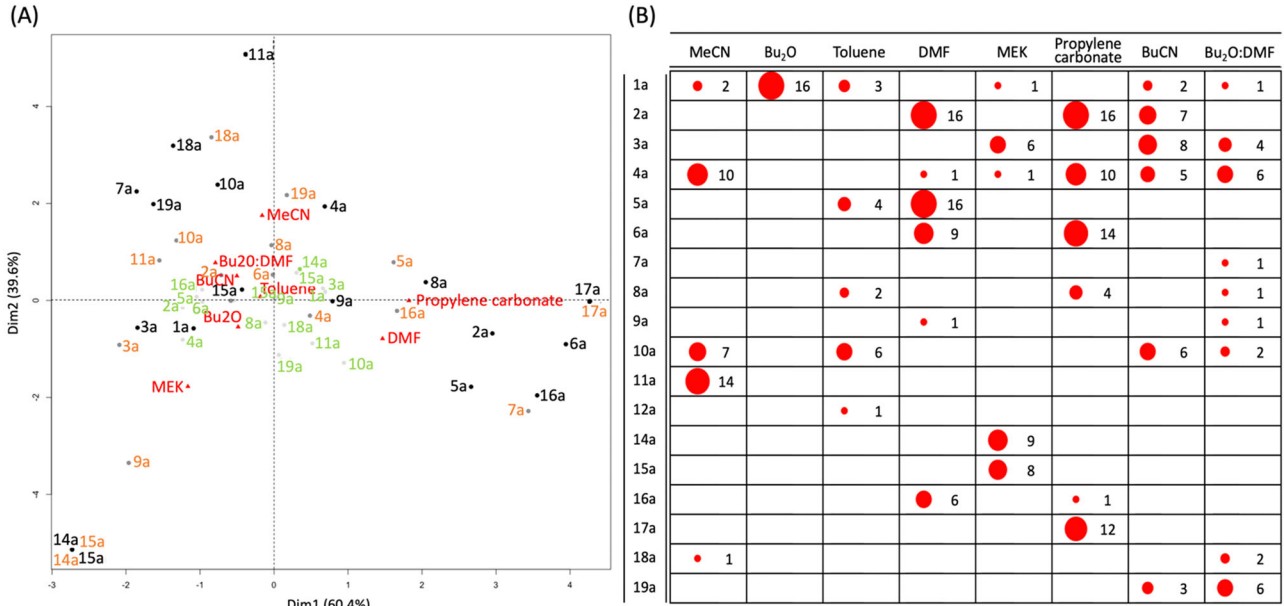

**Fig. 6 | Further data analysis. A** Correspondence analysis[74] biplot showing associations between solvents and reaction products. Key: Dim = dimension. The number of experiments (after combining replicate analyses) with high (black), medium (orange), and low (light green) quantities for each product are used in the analysis. **B** Bubble plot showing the number of experiments with quantities above two-thirds of the range for the product by solvent. Bubble sizes are proportional to the number of experiments, also shown where the maximum possible is 20 (i.e., 5 temperatures for 4 reaction times).

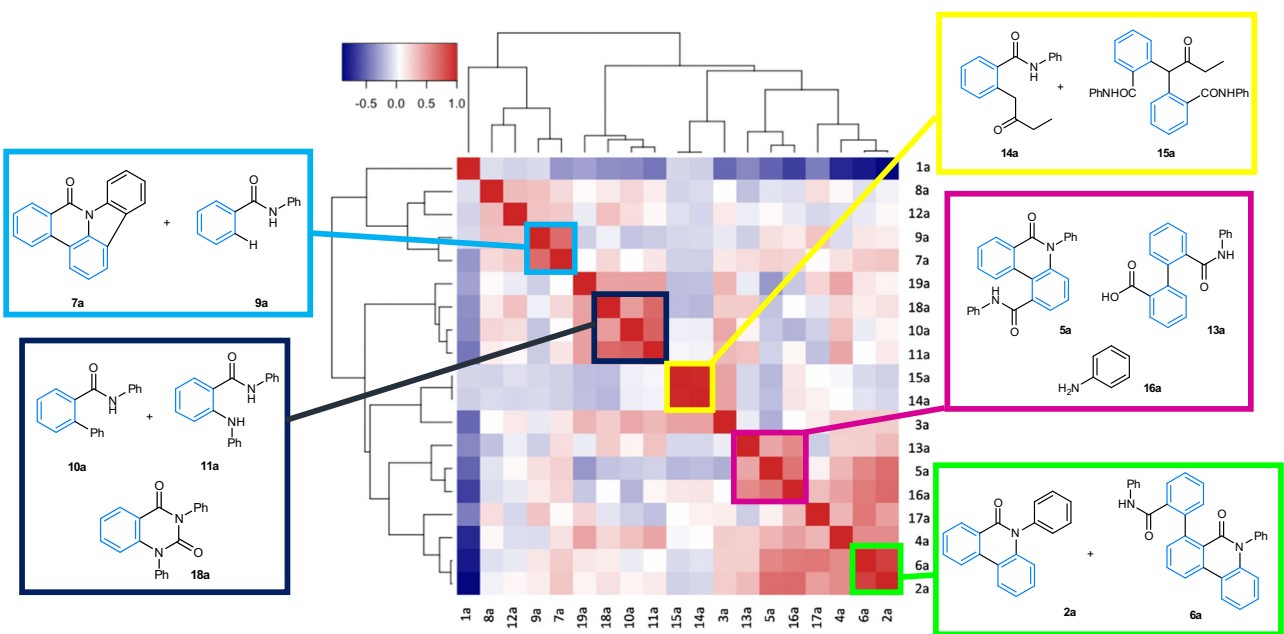

**Fig. 7 | Heatmap showing correlations between products across reactions, including all solvents, reaction times and temperatures.** Median values of replicate observations were used in the analysis. Key: red boxes are positive correlations and blue boxes are negative correlations (see color bar, upper left). The reaction products are ordered using hierarchical clustering, resulting in the dendrograms shown in the margins, so that similar products cluster together. The map colors show the strength of correlation, as indicated by the color bar. Specific clusters of compounds are highlighted by colored boxes, representing selected interactions.

product) grouped together. Compound **11a** (the Buchwald-Hartwig amination product) correlates with both **10a** (phenylated product) and quinazolinedione **18a**. Although not highlighted, Fig. 7 also shows the positive correlation of **5a** and **16a** with **2a**, supported by the similar loadings in PCA and their proximity in correspondence analysis (mainly due to being produced together in DMF reactions). Such data correlations form the basis for mechanistic proposals (vide infra).

## Further mechanistic analysis, supported by stoichiometric Pd chemistry experiments

To supplement our mechanistic predictions about the catalytic cycle(s) operative in this complicated chemistry, we recognized that our HTE/data analysis would be complemented by traditional stoichiometric studies involving appropriate Pd precursor compounds. The aim of this part of the study was not to provide definitive mechanistic information, but to support the connections made in the

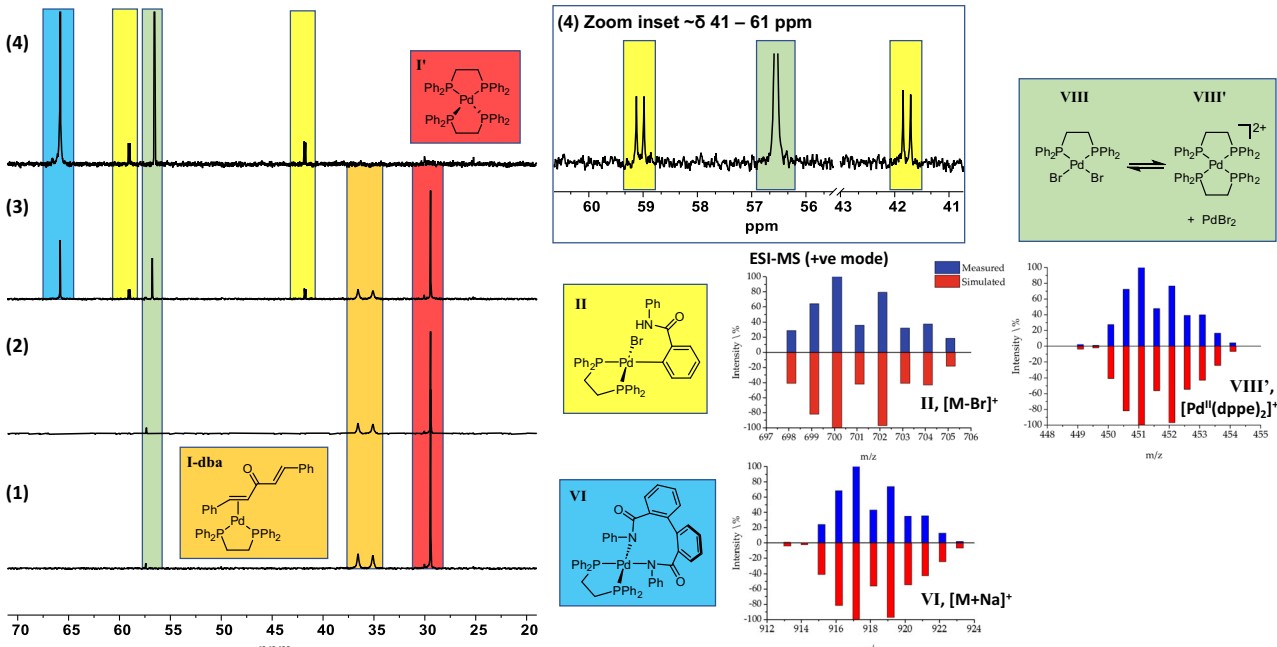

**Fig. 8 | Stoichiometric palladium reactivity studies using substrate 1a.** $^{31}$P NMR (203 MHz, DMF-$d_7$) spectral changes and confirmation of derived (pseudo)molecular ions by ESI-MS: (1) Pd$_2$dba$_3$·CHCl$_3$ and dppe taken at rt, $t = 0$ min; (2) taken after addition of 2-bromo-benzamide **1a** at rt, $t = 5$ min; (3) taken after 45 min heating at 80 °C; (4) taken after 16 h heating at 80 °C. We have assigned species to

five phosphorus species, cross-referenced with the ESI-MS analysis (simulated ions shown in red; experimental ions in blue). The species at δ 57.35 is a trace product, derived from the formation of Pd$^0$(dba)(dppe)/Pd$^0$(dppe)$_2$, which is distinct to the species at δ 56.77 ppm, formed at higher temperature, assigned to PdBr$_2$(dppe) **VIII**.

HTE/data analysis. Previously proposed catalytic cycles reported by ref. 20. were based primarily on DFT calculations using PH$_3$ as a model for the PPh$_3$ ligand (along with an N-methyl substrate variant of **1a**). Such a model makes a Pd$^{II}$/Pd$^{IV}$ catalytic cycle possible, in principle (Fig. 2A). While higher oxidation state Pd$^{IV}$ intermediates are experimentally feasible[24], evidence for the stabilization of Pd$^{IV}$ species by phosphines is relatively limited, exceptions being PPh$_3$[61] and transphos[62]. Indeed, the scientific community at large have often doubted Pd catalytic cycles involved phosphine-stabilized Pd$^{IV}$ intermediates[63,64]. While dppe could be related to these ligands, we conducted stoichiometric $^{31}$P NMR and MS experiments to better understand dppe interactions at Pd in the catalytic system of interest **1a** → **2a** and other side-products and by-products.

Amatore and Jutand showed that Pd$^0$(dppe)$_2$ **I'** and Pd$^0$(η$^2$-dba)(dppe) **I-dba** are formed from the reaction of Pd$_2$(dba)$_3$•dba with n dppe (n = 1 or 2) in THF[65]. In our hands, reaction of 1 equivalent of Pd(OAc)$_2$ with 1 equivalent of dppe in DMF-$d_7$ at 23 °C resulted in a mixture of Pd$^0$(dppe)$_2$ **I'** (δ 30.10 ppm; lit. δ 30.46 ppm in THF) and Pd$^{II}$(dppe)(OAc)$_2$ (δ 59.07 ppm; lit[66]. δ 58.9 ppm in CH$_2$Cl$_2$) being formed (Fig. 8). There was no evidence for phosphine oxidation or P-C bond cleavage (forming potential phosphido groups) in these experiments. As ligand dissociation from Pd$^0$(dppe)$_2$ **I'** is thought to generate the putative unsaturated (catalytic) Pd$^0$ species Pd$^0$(dppe) **I**, evidence was gained by addition of dba to the mixture of Pd$^{II}$(OAc)$_2$/1 dppe, and cross-referencing to an authentic sample of Pd$^0$(η$^2$-dba)(dppe) **I-dba** (δ 35.5 and 37.0 ppm, Δν$_{1/2}$ = 28 Hz)[65]. We also note that the presence of dba in the catalytic reaction does not significantly perturb the reaction system in terms of the many products that are formed (i.e., 0.5Pd$_2$(dba)$_3$/1dppe is a component catalyst system). The experiment confirmed Pd$^0$(dppe) liberation from Pd$^0$(dppe)$_2$ **I'** in DMF-$d_7$ at 23 °C; this latter species is an observable species, acting as a catalyst reservoir.

We assessed oxidative addition of 2-bromo-N-phenylbenzamide **1a** to a pre-synthesized mixture of Pd$^0$(dppe)$_2$ **I'** (Fig. 6, red) and Pd$^0$

(η$^2$-dba)(dppe) **I-dba** (Fig. 8, orange), derived from Pd$^0_2$(dba)$_3$•dba and 1 dppe in DMF-$d_7$ at 23 °C. It was necessary to heat the mixture of **1a** with these Pd$^0$ species at 80 °C to give key oxidative intermediate **II** (Fig. 6, yellow). Two doublets appeared at δ 41.78 and δ 59.05 ppm ($^2J_{PP}$ = 28 Hz), supporting formation of Pd$^{II}$(Ar)Br(dppe) **II**. The $^{31}$P NMR signals for oxidative addition products of this type are typically found in the δ 56–31 ppm range (in THF or DMF)[67,68]. In our case, the molecular fragment [M-Br]+ was verified by ESI-MS (+ve mode). A new singlet peak at δ 65.83 ppm (Fig. 8, blue) was observed, which we attribute to reaction intermediate **VI**, resulting from the formation of a C-C bond between two benzo-groups. This proposal is supported by the detection of the sodiated pseudo-molecular ion [**VI** + Na]$^+$ by ESI-MS (+ve mode).

Another new species with a signal of δ 56.55 ppm (Fig. 8, green) forms at 80 °C, which we tentatively assign to Pd$^{II}$(dppe)(Br)$_2$ **VIII** (Fig. 6, green). In a separate reported study an autoionization equilibrium was noted for the behavior of Pd$^{II}$(dppe)(OAc)$_2$ to give [Pd$^{II}$(dppe)$_2$]$^{2+}$ **VIII'** + Pd(OAc)$_2$[66]. The detection of [Pd$^{II}$(dppe)$_2$]$^{2+}$ **VIII'** by ESI-MS (+ve mode) allows us to assign this signal to Pd$^{II}$(dppe)(Br)$_2$, **VIII** with a similar autoionization equilibrium [Pd$^{II}$(dppe)$_2$]$^{2+}$ **VIII'** and PdBr$_2$ in play[66]. The formation of Pd black was noted at the end of this stoichiometric reaction, which is typical for these types of experiments.

Based on the complete information to hand we propose mechanistic hypotheses, as outlined in Fig. 9A. Firstly, for the catalyst activation step we have demonstrated that Pd$^0$(dppe) **I** species is generated under the reaction conditions, as the likely Pd$^0$ species in solution (in DMF), for which there are other potential reductants in the system (thus not dependent on the phosphine ligand per se, Fig. 9A). The generation of aniline **16a** under the working catalytic reaction conditions could assist formation of Pd$^0$, leading to the generation of dppe mono-oxide[69] (an alternative ligand, vide infra) and acetanilide **17a**, generated through acetate transfer to aniline **16a**. Given the higher reaction temperatures used for the catalytic reaction there are

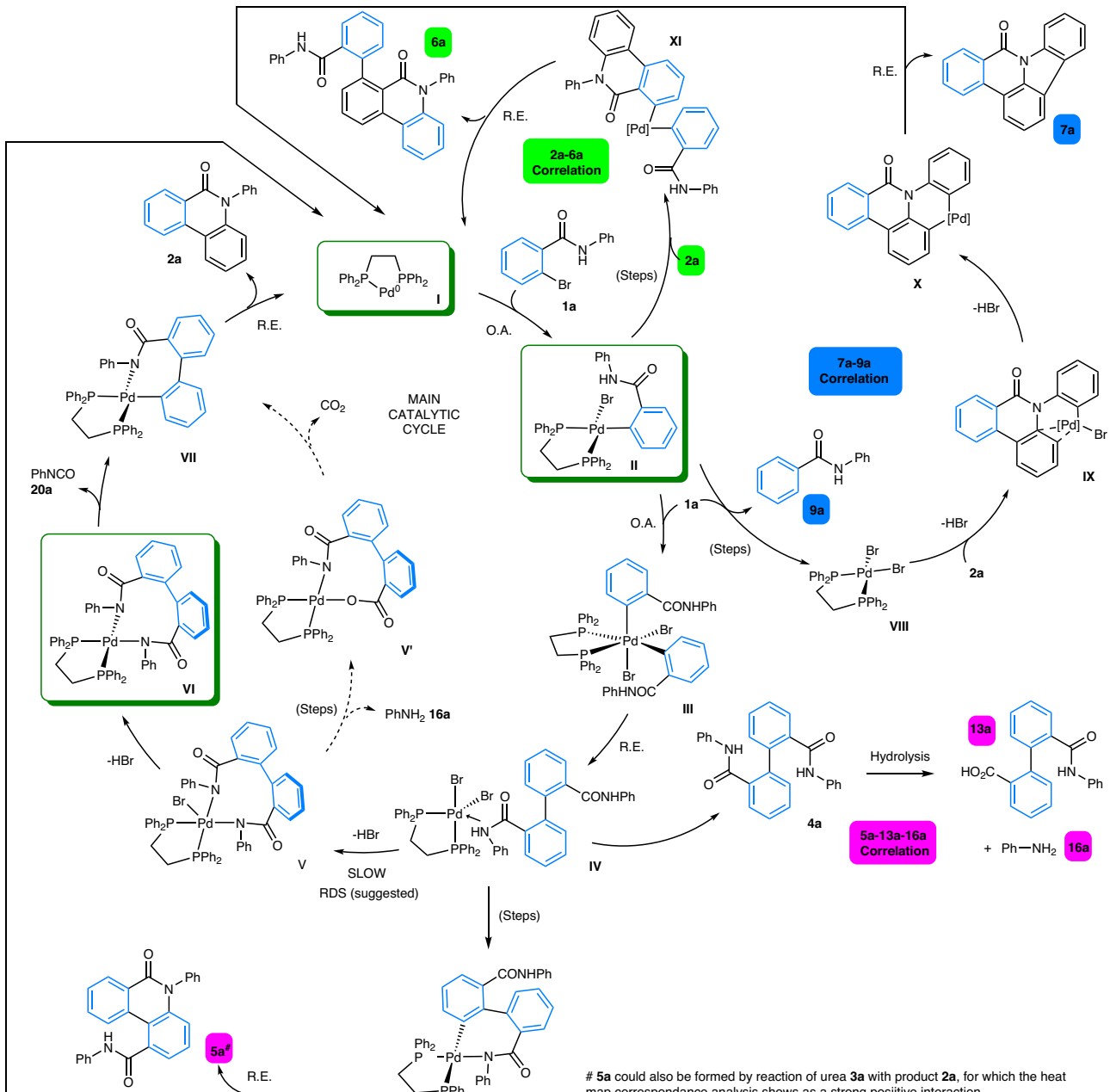

**Fig. 9 | Detailed catalytic cycle for the formation of phenanthridinone 2a.** Key: O.A. oxidative addition, R.E. reductive elimination, RDS is rate determining step. We expect all steps resulting in loss of HBr to involve base. Potential catalytic cycles for selected side-products, based on the correlations revealed by the rich data analysis (from heat maps and hierarchical clustering; highlighted by appropriate colors). We expect all steps resulting in loss of HBr to involve base. We do not preclude dppeO being an alternative ligand for Pd in these catalytic cycles, especially in solvents where there is no other potential reductant, i.e., where phosphine ligand becomes the obvious reductant. The blue color in the chemical structures shows the origin of the benzo-moiety.

different mechanisms possible for pre-catalyst activation and formation of Pd⁰ species (see Supplementary Information, Figure 15). We expect in polar aprotic solvents, e.g., DMF, that the presence of trace dimethylamine, CO and water would likely influence the reductive process.

We have confirmed experimentally that oxidative addition of 2-bromo-N-phenylbenzamide **1a** with Pd⁰ species containing dppe occurs to give Pdᴵᴵ species **II** containing new Pd-Br and Pd-C bonds (confirming scission of the C-Br bond in **1a**). From this point, a second oxidative addition-type reaction (mechanism not characterized) of 2-bromo-N-phenylbenzamide **1a** to **II** is necessary to generate a putative Pdᴵⱽ species **III** (following the mechanism[20] hypothesized by a Porée et al.). This is critical in generating a new C-C bond between the

required benzo-groups, which occurs by subsequent reductive elimination to give Pdᴵᴵ intermediate **IV**. The structure of **VI** is supported by NMR and MS, thus **IV** is connected by loss of two equivalents of HBr via **V** (which is likely base-assisted). This could in principle occur in a single step. We propose that **VI** is involved in the rate determining step/state (RDS). Extrusion of phenylisocyanate **20a** (which can be trapped by reaction with aniline **16a** to afford the urea **3a**) then affords Pdᴵᴵ intermediate **VII**, which we believe to possess a high energetic barrier. An alternative pathway from **V** → **VII** is shown (via **V′**), involving amide hydrolysis, formation of a Pd-O bond, extrusion of $CO_2$ and formation of aniline **16a**. The final step involves reductive elimination from **VII** to give phenanthridinone **2a** (the major and dominant product), with concomitant regeneration of the active Pd⁰ catalyst species "Pd⁰(dppe)" **I**.

Figure 9 further reveals how other major side-products **4a** (main cycle, de-coordination from Pd[II] intermediate **IV**), **5a, 6a,** and **7a** are connected to the main catalytic cycle. The heat map and hierarchical cluster analysis connecting **5a, 13a,** and **16a**, which are formed as downstream products from proposed intermediate **IV**, indicate that there are several exit points from the main catalytic cycle from which these side-products form.

We propose that pentacycle **7a** arises through a formal oxidative C-H activation of phenanthridinone **2a**. Thus, reaction of Pd(dppe)Br₂ **VIII** with **2a** would involve formation of Pd[II] intermediate **IX** (loss of HBr). Cyclopalladation, with loss of HBr[70] would give 6-membered ring palladacycle **X**. A classical reductive elimination step then forms **7a** and leads to the regeneration of 'Pd[0](dppe)' catalyst species **I**. The confirmed appearance of Pd(dppe)Br₂ **VIII** in our stoichiometric Pd experiments supports this potential oxidative process. The association of proto-debrominated product **9a** with the formation of **7a** (revealed by the Heat Map and hierarchical cluster analysis) suggests that 2-bromo-N-phenylbenzamide **1a** is capable of acting as an oxidant in this competing catalytic cycle. Organohalides acting as oxidants for Pd[0] → Pd[II] is established and has been exploited in other chemical synthesis applications[71,72]. We do not preclude a role for trace air in this process, serving a role to oxidize "Pd[0]" to "Pd[II]" species that would be equally capable of promoting the formation of **7a** through a similar sequence of steps. Qualitatively we can state that in the presence of air **7a** is formed in higher quantities.

The heat map correlation of side-product **6a** with **2a** allows us to propose that a C-H activation process involving phenanthridinone **2a** and Pd[II] oxidative addition intermediate **II** is likely, giving complex **XI** (detailed steps not given−cycle shown in Fig. 9). Subsequent reductive elimination regenerates the "Pd[0](dppe)" catalyst species **I**, releasing compound **6a** as another side-product. Catellani et al. has suggested that compounds similar to **6** could arise by intramolecular cyclopalladation, followed by coupling with **1a** and *ipso*-substitution[16]. We recognize that **6a** could derive directly from intermediate **VI**, through phenanthridinone ring-formation, retaining the phenyl amide group. This would account for **6a** having positive interactions with **2a** in the correlation.

Building on the discussion above about the correlation of **5a, 13a,** and **16a**−Porée et al. found[20] that compounds like **5** dominated under reaction conditions in which bromine−carbonate exchange and hence, rotation around the biaryl axis is disfavored. We acknowledge that **5a** could be formed by reaction of urea **3a** with product **2a**, for which the heat map analysis shows a strong positive correlation. Hydrolysis of **4a** likely occurs at Pd[II], thus carboxylic acid **13a** and aniline **16a** derived from intermediate **IV**.

The dppe ligand is a phenyl donor source that is transferred to **10a** and **12a**−it is established that phenyl-containing phosphines are capable of transferring a phenyl group at Pd[47]. On the other hand, hydrolysis product **8a** likely derives from oxidative addition intermediate **II**. The heat map correlation between **10a, 11a,** and **18a** shows that phenyl transfer from the dppe ligand is prevalent where amination occurs. Quinazolinedione **18a** formation is linked to **11a**, which we propose to be formed via a carbonylative process.

## Discussion

We have examined arguably one of the most complicated homogeneous-based Pd-catalyzed reactions reported in the literature – that is the reaction of 2-bromo-N-phenylbenzamide **1a** to give N-phenyl phenanthridinone **2a**, along with a plethora of by-products and side-products. We identified this reaction system as amenable to study better through the generation of a larger number of reaction outcomes, and profiling those reactions for all products formed in >0.1 %. Previous studies had highlighted the presence of some side-products and by-products, but the true extent of the reaction complexity was not known, moreover, the potential links between related and non-related products were not established[16−21]. Mechanisms had been hypothesized based on limited experimental evidence. Using automated high-throughput experimentation (HTE) methods and batch screening technologies, has enabled the effect of temperature and solvent on the reaction profile to be deduced. PCA of 480 batch reaction outcomes revealed the clear variance in starting material **1a**, product **2a,** and side-product profiles based upon solvent choice. Correspondence and Heat Map (with hierarchical clustering) analysis allowed us to confidently draw associations between reaction conditions and the interactions between different products. This detailed reaction profiling and statistical analysis approach has given us a better understanding of a complex reaction network and garnered new insight and chemical knowledge. Moreover, the approach we have devised can potentially serve as a blueprint for studying reaction complexity in other current and future reaction chemistries.

We can summarize our findings as follows: 1) The HTE reaction screening and associated data analysis revealed detailed information about the complexity associated with the Pd-catalyzed reaction of **1a** → **2a** and other side-products and by-products, involving many different types of bond activation and formation (i.e., C-Br, N-H and C-C). Post-functionalization of **2a** is affected by solvent polarity, reaction temperature and time. The profile and distribution of side-products indicate that there could be advantages in exploring both higher and lower reaction temperatures, and longer and shorter reaction times, to aid in promoting formation of one particular product (for discovery purposes). 2) Kinetic profiles revealed that **1a** is consumed over 1.5−2.25 h and that there is a confirmed pre-catalyst induction period (Fig. 2D), which can be reduced in time by pre-heating the mixture of Pd(OAc)₂ and dppe. This approach is not commonly applied in synthetic chemistry, and we recommend that both regimes are tested in assessing the full-scope of a metal-catalyzed reaction. 3) The Pd pre-catalyst activation pathway has been examined stoichiometrically in DMF solvent, which confirmed that Pd[0](dppe)ₙ species are formed under the reaction conditions tested. While the mono-oxide of dppeO was not detected in this reaction, a potential role for this alternative hemilabile bidentate ligand is not discounted in other reaction solvents, particularly under catalytically-relevant conditions and/or where trace air is in the system. Practically speaking, the phosphine can be a sacrificial reductant for Pd(II) to Pd(0), particularly for reactions employing Pd(OAc)₂. The use of Pd₂(dba)₃/dppe can avoid this issue, so long as O₂ is minimized in the system. Furthermore, it is important to consider that the original ligands added to a catalytic reaction involving a transition metal might be different under working reaction conditions. Any structural modification could alter the ligand steric and electronic effects. For a bidentate ligand, the bite angle and hemilability could also be changed. 4) The heat map and hierarchal clustering correlations were evidenced by − a) Concomitant formation of pentacycle **7a** with proto-debrominated benzamide **9a** allows us to conclude that **1a** is a potential oxidant[71,72] in this competing catalytic cycle (note: we do not preclude the presence of trace air-promoting formation of **7a** via oxidic Pd[II] species); b) Positive interactions between **5a, 13a,** and **16a** confirm that there is an exit point from the main catalytic cycle involving species **IV**; c) Correlation of side-product **6a** with **2a** allows us to propose that a C-H activation process involving phenanthridinone **2a** and Pd[II] oxidative addition intermediate **II** is possible. However, it is feasible that **6a** could derive from intermediate **VI**, thus **6a** is formed along with **2a**. 5) Complementary stoichiometric organopalladium studies has allowed us to gain insights into likely reaction Pd intermediates, particularly an oxidative addition intermediate and advanced downstream Pd[II] intermediate following activation of two molecules of 2-bromo-N-phenylbenzamide **1a**. There are benefits to be gained from such experiments in understanding what are the likely catalytic species and catalyst resting states, i.e., prior to conducting computational studies and more in-depth kinetic studies. 6) The rate determining step is proposed to be extrusion of phenyl

isocyanate **20a**, affording intermediate **VII**, based on the appearance of **VI** experimentally. To have an indication about the rate determining step can potentially aid future reaction development, particularly how this might change under varying reaction conditions.

The combined HTE, data rich analytical approach, complemented by traditional organopalladium mechanistic studies, has allowed us to gain new insight into a highly complicated Pd-mediated processes. We expect our approach in general terms, and the techniques employed, could find wide application in the field of transition metal catalysis and applied synthetic chemistry. Indeed, the full mapping out of the complexity of the reaction system (as described above), with a focus on the identification and distribution of side products and by-products, could be both a useful and rewarding point to optimize to. Moreover, profiling more broadly the complete fingerprint of a given reaction system could contribute towards a more holistic understanding of chemical reaction Tox-Profiles, as outlined by ref. 73.

## Methods

### Synthetic procedure for the major product, 5-phenyl-5,6-dihydrophenanthridin-6-one 2a

$Pd(OAc)_2$ (5.6 mg, 0.025 mmol, 5 mol%), dppe (10.0 mg, 0.025 mmol, 5 mol%), and $K_2CO_3$ (138 mg, 1.0 mmol, 2.0 eq.) were placed in a Schlenk tube under $N_2$. DMF (1.5 mL) was added, and the mixture was stirred at 80 °C for 2 min. Then, a solution of 2-bromo-N-phenylbenzamide **1a** (138 mg, 0.5 mmol, 1.0 eq.) in DMF (1 mL) was added via cannula. [Air was injected into the reaction mixture via syringe (5 mL)—to assess the impact of trace air in the reaction systems]. The reaction mixture was then stirred at 80 °C for 6 h then allowed to cool to room temperature (~22 °C). Then, EtOAc (10 mL) was added, and the resulting mixture was filtered through Celite and washed with EtOAc (3×5 mL). The combined organic extracts were was washed with 1 M HCl(aq) (10 mL) and brine (10 mL), dried over $MgSO_4$, filtered, and the concentrated under reduced pressure to give a crude product which was subsequently purified by flash column chromatography ($SiO_2$, 9:1 to 8:2 petrol:EtOAc), to afford compound **2a** as a white solid (39 mg, 58%).

## Data availability

The raw data from the HTE screening experiments (LC-MS data) has been deposited to Research Data York (freely available). See DOI: 10.15124/37869e97-c94a-41a7-9f69-fe3ed1c10984. Representative NMR spectral data (included in the Supplementary Information document) can also be found in the same data location. Source data is also provided as a single.csv file (Supplementary Information). The single crystal X-ray diffraction files (.cif) have been deposited to the Cambridge Crystallographic Database (compound **2a**, CCDC 2063167; compound **2b**, CCDC 2063164; compound **4a**, CCDC 2063166); compound **5a**, CCDC 2081723; compound **7a**, CCDC 2063165; compound **S3** (supplementary reference compound), CCDC 2081724). All data are available from the corresponding authors. Source data are provided with this paper.

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

## Acknowledgements

We are grateful to Chemspeed Technologies Ltd to a Partnership with the University of York which led to the embedment of an ISYNTH robotic platform within our laboratories, with particular thanks to Christian Dittrich, Jake Grace, Stephane Rickling, and Andrew Stephenson for supporting our efforts with technical training and maintenance. We have been principally funded by the EPRSC for this research (EP/S009965/1; "A Fully-Automated Robotic System for Intelligent Chemical Reaction Screening") and supported by EPSRC IAA awards and a Centre for Future Health ("CFH1 Partnership—Pharmaceutical Optimization using a Laboratory Automated Reaction Intelligent System (POLARIS)") by the University of York. The research leading to these results has received funding from the Innovative Medicines Initiative Joint Undertaking under grant agreement no. 115360, resources of which are composed of financial contributions from the European Union's Seventh Framework Programme (FP7/2007–2013) and EFPIA companies' in-kind contributions (to L.A.L., D.J.M. and I.J.S.F.). The research was supported in part by another EPSRC grant, with follow-on funding for C.S.H. and J.B.E. (EP/W031914/1). I.J.S.F. and J.M.L. are currently supported by Royal Society Industry Fellowships. R.A.B. was supported by the Royal Academy of Engineering under the Research Chairs and Senior Research Fellowships scheme.

## Author contributions

I.J.S.F. conceived and designed the original research project examining the underpinning mechanistic chemistry. L.A.L. designed and executed these initial synthetic and mechanistic experiments (PhD study), directed by I.J.S.F. with co-supervision provided by D.J.M. L.A.L. was supported by work conducted by R.C. and A.P. J.B.E. conducted the in operando IR spectroscopic analyses. A high throughput screening and data analysis project laid the foundations of GEC's PhD project, which was designed by I.J.S.F., J.M.L., J.M.S. and J.W., who all played a role in the supervision of G.E.C. Synthetic chemistry experiments, HTE, and other data analysis were conducted as part of G.E.C.'s PhD studies (major part of thesis); P.L.M. conducted supporting experiments along with G.E.C. J.T.W.B. conducted screening experiments on the Chemspeed ISYNTH system, helping to direct and guide G.E.C. Both C.S.H. and R.A.B. assisted with the design and execution of automated experiments. A.C.W., J.M. and S.H. conducted single-crystal X-ray diffraction experiments, refined data, and solved the chemical structures described in the manuscript. J.D.F. (re)synthesized the majority of organic compounds described in the manuscript. J.D.F. wrote a first draft of the manuscript and Supplementary Information documents, in consultation with I.J.S.F. and J.W. This involved assembling data from PhD theses (L.A.L. and G.E.C.), cross-referenced with other works conducted in the laboratory by others. I.J.S.F. and J.W. co-led the writing of the final manuscript, including revision and creation of figures. The experiments leading to the results described in this manuscript were conducted locally at the University of York. The project involved a collaboration (and funded research projects) with the University of Leeds (a local partner), which aided study design and implementation. Further experiments were conducted at the University of Leeds on the use of continuous flow reaction screening (led by R.B.). Those results will be reported separately. The research was not restricted or prohibited in the setting of the researchers. It was not necessary to seek local ethics approval for the program of research that led to the results described in this manuscript. Local and national guidelines were followed concerning the use of potentially hazardous substances, for which appropriate individual experimental risk assessments were made.

## Competing interests

I.J.S.F. declares a potential competing interest associated with this research. All other authors do not have competing interests. Working in partnership with Chemspeed Technologies, ISYNTH equipment was embedded within the Chemistry laboratories in York in 2012. A contract with the University of York was put in place for the Chemspeed Technologies UK to be based in York from 2012–19. All research conducted on the equipment was free from direct involvement with the company. The research did not require prior approval for publication of this research. The company checked the acknowledgements section prior to submission of the manuscript for peer review.
