## [Peer Review File · Nature Communications]

REVIEWER COMMENTS

Reviewer #3 (Remarks to the Author):

This revised article by Fairlamb et al. addresses many of the specific concerns presented by the other reviewers. I was reviewer 3 on the previous version, and my primary concern was not necessarily about the content, but rather the presentation of the content (i.e., that it was dense, detailed, somewhat disjointed, and lacking a clearly stated justification for why the work matters - therefore perhaps not appropriate for a broad readership). This concern still stands, but the authors have added a portion to the conclusions that summarizes the key take-away points (which I think increases the likelihood that an average reader would understand the gist of this paper).

To clarify my perspective, I really admire this work - it is an impressive endeavor and of high scientific quality. I am personally someone who easily gets caught up in the minutia of my own research problems and loves the concept of leaving no stone unturned in understanding mechanism. The authors have done exactly that. I am advocating for this manuscript to be published BUT I want the authors to tweak it so that it will land better with their target audience. The following describes my thoughts on how to do this.

In my previous review I commented that the work "doesn't really lead to substantial new insight". Reviewer 1 arrived at a similar conclusion ("the whole does not appear to be better than the sum of the parts"). The additions to the Conclusion section in this version of the manuscript are helpful in addressing this concern, but I think there is still significant room for improvement in this regard.

In the response to reviewers, the authors argue that there in fact is "substantial new insight" and that it is "the hidden features of what is arguably the most complicated Pd-catalysed reaction known in the chemical literature". I appreciate that. But why would someone want to know the hidden features of any reaction? For some people the answer may be simple intellectual curiosity - knowing for the sake of knowing, and I think that's what the authors may be trying to argue - but I think that a large portion of scientists want to understand mechanism so that they can exploit the understanding

to make improvements, develop new reactions, etc. (Certainly this is why funding agencies or for-profit businesses like pharma would care about mechanistic insight.)

I feel this manuscript comes just short of taking that final step of connecting the new mechanistic insight with practical implications. Can the authors figure out a sentence they could add to (ideally) each bullet point of the conclusions, to comment on how each piece of mechanistic insight is valuable? For example, the first bullet point mentions side products and byproducts... can the authors comment on how the mechanistic insight suggests changes that could be made to minimize one or more side/byproducts, depending on which one(s) a user finds most problematic? The second bullet point is the only one that alludes to a practical implication (i.e., how to shorten the induction period), so this is good. The third bullet point discusses precatalyst activation pathways: are there any practical implications from this mechanistic insight? Does it suggest what kind of conditions should be avoided, or how one might consider changing their precatalyst if they need a faster reaction/lower catalyst loading? Etc.

The authors could also consider whether it would be appropriate to mention the practical implications of the mechanistic insight (perhaps in more detail) earlier in the manuscript as well, though I would be happy just seeing these in the conclusions. I acknowledge that the authors do this already in a few places (e.g., they imply that product 11a could be decreased by moving away from MeCN as the solvent), but those implications get a little lost in the main text and aren't always stated explicitly.

Reference NCHEM-23040761: Response to Reviewer Comments

We have distilled out the key questions and suggestions, responding to these below, which are accompanied by the highlighted changes in the manuscript.

Q1 Can the authors comment on how the mechanistic insight suggests changes that could be made to minimize one or more side/byproducts, depending on which one(s) a user finds most problematic?

Q1 Reply We have included a comment about the solvent type, reaction temperature and reaction time. Critically, we wanted to ensure that the reader understood that one can go to higher and lower temperatures, and longer and shorter reaction times to explore the complete reaction space. One does not typically see this done in reaction screening, as the motivation is often to find the mildest (lowest energy demand) reaction conditions.

Q2 The second bullet point is the only one that alludes to a practical implication (i.e., how to shorten the induction period), so this is good.

Q2 Reply Thank you. We have developed this point further, as synthetic chemists generally do not pre-heat ligand and pre-catalysts – a typical approach is mix everything together, and we see that in many catalytic synthetic methodologies reported in recent years.

Q3 The third bullet point discusses precatalyst activation pathways: are there any practical implications from this mechanistic insight? Does it suggest what kind of conditions should be avoided, or how one might consider changing their precatalyst if they need a faster reaction/lower catalyst loading? Etc.

Q3 Reply We have added a few further comments to strengthen this point. Rigorously excluding oxygen in a reductive-type Pd-catalyzed cross-coupling is clearly recommended, although we suspect not practical for most synthetic chemistry laboratories. We have made a practical suggestion about the use of $\text{Pd}_2(\text{dba})_3$ instead of 'Pd(OAc)₂'. For the latter the phosphine ligand is a good sacrificial reductant for Pd(II) to Pd(0).

Suggestion: The authors could also consider whether it would be appropriate to mention the practical implications of the mechanistic insight (perhaps in more detail) earlier in the manuscript as well, though I would be happy just seeing these in the conclusions. I acknowledge that the authors do this already in a few places (e.g., they imply that product 11a could be decreased by moving away from MeCN as the solvent), but those implications get a little lost in the main text and aren't always stated explicitly.

Reply: We did consider this suggestion but felt that enough mechanistic details are discussed in the main part of the manuscript. The conclusions section is now significantly strengthened, according to the reviewer's excellent suggestions.

Original reviewer comments (key points highlighted which are addressed above).

Reviewer #3 (Remarks to the Author):

This revised article by Fairlamb et al. addresses many of the specific concerns presented by the other reviewers. I was reviewer 3 on the previous version, and my primary concern was not necessarily about the content, but rather the presentation of the content (i.e., that it was dense, detailed, somewhat disjointed, and lacking a clearly stated justification for why the work matters - therefore perhaps not appropriate for a broad readership). This concern still stands, but the authors have added a portion to the conclusions that summarizes the key take-away points (which I think increases the likelihood that an average reader would understand the gist of this paper).

To clarify my perspective, I really admire this work - it is an impressive endeavor and of high scientific quality. I am personally someone who easily gets caught up in the minutia of my own research problems and loves the concept of leaving no stone unturned in understanding mechanism. The authors have done exactly that. I am advocating for this manuscript to be published BUT I want the authors to tweak it so that it will land better with their target audience. The following describes my thoughts on how to do this.

In my previous review I commented that the work "doesn't really lead to substantial new insight". Reviewer 1 arrived at a similar conclusion ("the whole does not appear to be better than the sum of the parts"). The additions to the Conclusion section in this version of the manuscript are helpful in addressing this concern, but I think there is still significant room for improvement in this regard.

In the response to reviewers, the authors argue that there in fact is "substantial new insight" and that it is "the hidden features of what is arguably the most complicated Pd-catalysed reaction known in the chemical literature". I appreciate that. But why would someone want to know the hidden features of any reaction? For some people the answer may be simple intellectual curiosity - knowing for the sake of knowing, and I think that's what the authors may be trying to argue - but I think that a large portion of scientists want to understand mechanism so that they can exploit the understanding to make improvements, develop new reactions, etc. (Certainly this is why funding agencies or for-profit businesses like pharma would care about mechanistic insight.)

I feel this manuscript comes just short of taking that final step of connecting the new mechanistic insight with practical implications. Can the authors figure out a sentence they could add to (ideally) each bullet point of the conclusions, to comment on how each piece of mechanistic insight is valuable? For example, the first bullet point mentions side products and byproducts... Q1 can the authors comment on how the mechanistic insight suggests changes that could be made to minimize one or more side/byproducts, depending on which one(s) a user finds most problematic? Q2 The second bullet point is the only one that alludes to a practical implication (i.e., how to shorten the induction period), so this is good. Q3 The third bullet point discusses precatalyst activation pathways: are there any practical implications from this mechanistic insight? Does it suggest what kind of conditions should be avoided, or how one might consider changing their precatalyst if they need a faster reaction/lower catalyst loading? Etc.

Suggestion The authors could also consider whether it would be appropriate to mention the practical implications of the mechanistic insight (perhaps in more detail) earlier in the manuscript as well, though I would be happy just seeing these in the conclusions. I acknowledge that the authors do this already in a few places (e.g., they imply that product 11a could be decreased by moving away from MeCN as the solvent), but those implications get a little lost in the main text and aren't always stated explicitly.